# Optimizing Volumetric Ratio and Supporting Electrolyte of Tiron-A/Tungstosilicic Acid Derived Redox Flow Battery

**DOI:** 10.3390/ma18194614

**Published:** 2025-10-05

**Authors:** Yong Jin Cho, Jun-Hee Jeong, Byeong Wan Kwon

**Affiliations:** Department of Energy Chemical Engineering, Kangwon National University, Samcheok 25913, Republic of Korea; whdydwls6740@gmail.com (Y.J.C.); junhi1227@kangwon.ac.kr (J.-H.J.)

**Keywords:** aqueous redox flow battery, volumetric optimization, tungstosilicic acid, activated tiron

## Abstract

Redox flow batteries (RFBs) are a promising technology for large-scale energy storage due to their safety, scalability, and design flexibility. This study investigated a tiron-A (4,5-dihydroxybenzene-1,3-disulfonic acid)/tungstosilicic acid (TSA) RFB system, focusing on optimizing the supporting electrolyte and the volumetric ratio of the catholyte (tiron-A) to anolyte (TSA). Electrochemical characteristics, confirmed by CV and EIS, showed that sulfuric acid was the most suitable supporting electrolyte due to its excellent cell potential and lower ohmic resistance compared to sodium chloride and sodium hydroxide electrolytes. To address the inherent electron capacity imbalance between tiron-A (two electrons) and TSA (four electrons), various volumetric ratios were evaluated. The cell with the 3:1 tiron-A:TSA ratio exhibited optimal performance, achieving the highest discharge capacity, excellent cycle stability, and consistent energy efficiency. The electrochemical impedance spectroscopy results revealed that the ohmic resistance was minimized at the 3:1 ratio. This stable, low-ohmic resistance, coupled with a significant reduction in charge transfer resistance after cycling, was confirmed as the dominant factor for the improved long-term performance. These findings demonstrate an effective strategy for developing a high-performance performance tiron-A/TSA RFB system.

## 1. Introduction

Population growth and technological development have led to increasing energy consumption [1]. Simultaneously, electricity generation derived from conventional fossil fuels has been accelerating significant environmental problems. Therefore, new renewable energy technologies have been developed to alleviate this acceleration. These days, wind power and solar power generation systems occupy large portions of the renewable energy sources [2]. However, these are limited by geographical and weather conditions. To solve this limitation, an energy storage system (ESS) has gained attention. The ESS can store energy through chemical, physical, or hybrid methods, allowing for affordable load leveling and reliable energy supply [3,4]. Among various ESS technologies, redox flow batteries (RFBs) are particularly promising due to their safety, scalability, design flexibility, and capacity control [5]. These advantages make RFBs commonly utilized for large-scale ESS [6]. The RFBs store energy through reversible redox reactions between anolyte and catholyte [7]. Currently, the all-vanadium RFB (VRFB) has been most widely utilized in power plant industries [8]. Nevertheless, the VRFB system suffers from several drawbacks, including low energy density (25–30 Wh/kg), a narrow operating temperature range (10–40 °C), and a high cost (USD330–880 kW/h) [9,10,11]. Hence, a number of alternatives have been investigated instead of the vanadium-based materials. Polyoxometalates (POMs), which are heavy-weight metal-oxygen clusters, have emerged as promising candidates due to their multi-redox properties, high solubility, and relatively large energy density [12,13]. On the other hand, POMs also have limitations regarding their non-reversible properties in certain conditions [14]. Therefore, a lot of studies have focused on enhancing POM-based RFB by optimizing the cell conditions, including material composition, solvent, solubility, viscosity, pH, and additives. Especially, the capacity and efficiency of RFBs are critically dependent on the type and concentration of the supporting electrolyte, and volumetric conditions [15,16,17]. These various factors must be carefully controlled when designing the POM-based RFB system.

This study focused on optimizing the supporting electrolyte and volumetric control factors for tiron-A (tiron-activation, 4,5-dihydroxybenzene-1,3-disulfonic acid)/tungstosilicic acid (TSA) RFB system. Finding a suitable supporting electrolyte is fundamental and crucial for RFB system design. Volumetric control is another critical factor that directly influences cell performance. D. Mu Et Al. (2017) addressed a strategy for alleviating capacity fading by adding extra anolyte in VRFB [18]. According to the study, adding the +10% anolyte exhibited optimum performance, with +20, and +30% addition also improving capacity retention. J. Shin Et Al. (2024) investigated the relationship between electrolyte volume imbalance and the concentration of the supporting electrolyte [19]. They found that adjusting the catholyte concentration to 2.0 M of KOH enhanced the water diffusion and long-term stability. P. Navalpotro Et Al. (2025) deeply studied the interaction between the supporting electrolyte and redox-active species [20]. The study introduced various factors, including electrostatic, chemical, physical, interfacial, transport, and steric interactions for the organic RFB system. Especially, the study pointed out that the selection of the supporting electrolyte directly affects cell performance. The volumetric control is a common technique used to mitigate crossover phenomena by balancing the reaction rates of the two half-cells [21,22]. Thus, optimizing cell performance can be achieved by determining the appropriate volumetric ratio of the anolyte and catholyte. In this study, the energy densities of the anolyte (TSA) and catholyte (tiron-A) are inherently different, since the TSA redox reaction involves four electrons while the tiron-A reaction involves two. The TSA, Keggin-type polyoxometalate, is highly soluble in water, exhibits multiple redox reactions, and maintains structural stability [14,23]. The tiron, disodium 4,5-dihydroxybenzene-1,3-disulfonate, which belongs to the benzoquinone family, has been previously studied as a cathodic material due to its high solubility in acidic media and high redox potential. To prepare the tiron-A, an ion exchange resin (Amberlyst^®^ 15) treatment was used to remove sodium ions [11]. This transformation mechanism is illustrated in Figure 1 [24]. Thereby, this study newly arranges and evaluates the tiron-A/TSA RFB system.

## 2. Experimental

### 2.1. Materials

Tungstosilicic acid was used for anolyte active species, and it was prepared from tungstosilicic acid hydrate (H_4_(SiW_12_O_40_)∙H_2_O, Aladdin Scientific, Shanghai, China). The tiron-A (4,5-dihydroxy-1,3-benzenedisulfonic acid) was used for catholyte active species, and it was prepared from 4,5-dihydroxy-1,3-benzenedisulfonic acid disodium salt monohydrate ((OH)_2_C_6_H_2_(SO_3_Na)_2_∙H_2_O, Sigma-Aldrich, St. Louis, MO, USA). The representative alkaline, neutral, and acidic supporting electrolyte candidates are sodium hydroxide (NaOH, Alfa Aesar, Haverhill, MA, USA), sodium chloride (NaCl, Alfa Aesar, Haverhill, MA, USA), and sulfuric acid [H_2_SO_4_, Sigma-Aldrich]. They were dissolved into the deionized water (DI water) with a concentration of 1.0 M. Nafion^TM^ N-115 (Chemours Company, Wilmington, DE, USA) membrane was used to transfer the protons between anolyte and catholyte during Charge–Discharge cycling.

### 2.2. Preparation

To remove the sodium ions from the sulfonate, the tiron solution was passed through the Amberlyst^®^ 15 ion exchange resin (hydrogen form, Sigma-Aldrich). The resin was first fixed in a burette and then washed once with a 0.5 M sulfuric acid solution. Subsequently, the tiron solution was poured onto the washed resin. Finally, the tiron-A solution was obtained. The N-115 membrane was pretreated to improve its ionic conductivity using the following sequential method: (1) immersion in 5 wt% hydrogen peroxide for 1 h at 80 °C, (2) washing with DI water for 1 h at 80 °C, (3) immersion in 0.5 M of sulfuric acid for 1 h at 80 °C, and (4) a final wash with DI water for 1 h at 80 °C.

### 2.3. Electrochemical Analysis

A three-electrode system was employed to evaluate the electrochemical properties of the anolyte and catholyte. This system consisted of a working electrode (glassy carbon, 0.0707 cm^2^), reference electrode (Ag/AgCl, 3.0 M of KCl), and counter electrode (Pt wire). Cyclic voltammetry (CV) was performed using an SP-50 potentiostat (Biologic, Seyssinet-Pariset, France) to analyze the specific redox properties of the electrolytes. The specific potential range was 0.0–1.5 V for tiron-A, and −0.7–0.3 V for TSA with a scan rate of 100 mV/s. Electrochemical impedance spectroscopy (EIS) was conducted using a ZIVE SP-1 instrument (WonATech, Seoul, Korea) to investigate their impedance. A two-electrode system was utilized to analyze the unit cell properties via the EIS technique. The electrodes were directly connected to the current collectors of the unit cell. The specific frequency range was 100,000–0.1 Hz for both two- and three-electrode systems, while the amplitude was 10 mV.

### 2.4. Battery Performances

The RFB unit cell was constructed from a series of components: two end plates, two current collectors (gold-plated), two graphite plates, two cell frames, two electrodes (carbon felt, 6 cm^2^), and a pretreated N-115 membrane. A WBCS3000 cycler (WonATech) was used to evaluate the capacity and efficiency of the cell. During charge–discharge cycles, the cell was operated at a constant current of 360 mA with a voltage window of 0.0 V to 1.3 V. The experimental setup is illustrated in Figure 2.

## 3. Results and Discussion

### 3.1. Supporting Electrolyte

Figure 3 shows the cyclic voltammogram (CV) and Nyquist plot results for the tiron-A and TSA in three different supporting electrolytes: sodium hydroxide (alkaline), sodium chloride (neutral), and sulfuric acid (acidic). In the sodium hydroxide electrolyte, both active materials exhibited degradations, resulting in significantly reduced redox reactions and lower current densities. In addition, the reduction peaks of tiron-A were not observed. Hence, the sodium hydroxide is clearly unsuitable as the supporting electrolyte material for this system. In contrast, both the sodium chloride and sulfuric acid electrolytes showed clear and distinct redox peaks for both TSA and tiron-A. The specific redox potential values and their reaction mechanisms are summarized in Table 1 [24,25]. To understand the redox reaction of TSA, the pH of the initial solution was measured to determine the characteristics of its ionized form. The pH of a 0.1 M TSA solution was measured to be 0.50, corresponding to an approximate proton concentration of 0.316 M [26]. This value implies the partial dissociation of TSA as H(SiW_12_O_40_)^3−^ and three protons (H^+^). Therefore, the redox reaction occurred three times as shown in Table 1. These measured potential values correspond to the respective oxidation and reduction reaction voltages. To elaborate, the measured potential values were converted to the reversible hydrogen electrode (RHE) potential using Equation (1) [27]. This conversion allows us to compare unified potential as the pH values differ between sodium chloride and sulfuric acid supporting electrolytes. Their specific pH values are provided in Table 2.(1)$$E_{RHE}=E_{Ag/AgCl}+0.059\;pH+0.198\;V$$

The 0.198 V in the equation indicates the potential difference between the hydrogen electrode and Ag/AgCl reference potential. According to the potential values, sulfuric acid supporting electrolyte produced a narrower oxidation-reduction potential gap compared to the sodium chloride supporting electrolyte. Furthermore, the electromotive force (EMF), or the potential gap between TSA anolyte and tiron-A catholyte, was measured to be 0.795 V for the sodium chloride electrolyte and 0.860 V for the sulfuric acid electrolyte. This potential gap represents the overall cell potential [28]. Figure 3d show the Nyquist plots for each of the electrolytes. For evaluating the electrolyte, the ohmic resistance, which represents the initial resistance caused by the electrolyte, is particularly important. This value commonly corresponds to the x-intercept of the Nyquist plot [29]. According to the results, the 1.0 M sulfuric acid electrolyte exhibited the lowest ohmic resistance, making it the most suitable electrolyte. Additionally, based on the CV results discussed earlier, the use of the sulfuric acid electrolyte also yielded the highest cell potential. Figure 4 presents additional CV results with extended scan rates. The oxidation (oxi.) and reduction (Red.) are separately presented. The Randles-Sevcik equation is used to calculate the specific diffusion coefficient [30]. Accordingly, the diffusion coefficient can be calculated from Equation (3).(2)$$i_{peak}=2.69\times 10^{5}n^{\frac{3}{2}}AD^{\frac{1}{2}}Cv^{\frac{1}{2}}$$(3)$$D={\left(\frac{m}{2.69\times 10^{5}n^{\frac{3}{2}}AC}\right)}^{2}$$
where *i_peak_* is the peak current, *n* is the number of electrons in the electrochemical reaction, *A* is the surface area of the working electrode, *D* is the diffusion coefficient, *C* is the concentration of the active species, *v* is the scan rate, and the *m* is the slope of the *i_peak_* vs. *v*^1/2^ plot. The measured diffusion coefficients are provided in Table 3. According to the results, a greater diffusion coefficient was observed in the sulfuric acid supporting electrolyte than in the sodium chloride supporting electrolyte. Figure 4 also provides reversibility of the redox reactions in the two electrolytes. The shift in peak potential (*E_p_*) with varying scan rate indicates that the redox reaction is quasi-reversible or irreversible [31]. The increasing slope (*m*) of the *E_p_* vs. log *v* plot indicates a corresponding increase in the irreversibility of the redox reaction [32]. The results show that the slopes of the redox reactions, except for the third oxidation peak in the TSA solution, are lower when the sulfuric acid supporting electrolyte is used compared to the sodium chloride electrolyte. Therefore, based on higher cell potential, lower ohmic resistance, greater diffusion coefficient, and improved electrochemical reversibility, the sulfuric acid supporting electrolyte is highly suitable for the tiron-A/TSA RFB system.

### 3.2. Cell Performances

Figure 5 displays the performance of the unit cells corresponding to the volumetric ratio of the tiron-A:TSA electrolyte. The 1:1 ratio cell serves as a baseline. According to Figure 5a,b, the TSA-rich cells (1:2 and 1:3 ratios) showed reduced discharge capacity and exhibited gradual capacity fading over cycles. In contrast, the tiron-A-rich cells (2:1, 3:1, and 4:1 ratios) exhibited a higher capacity than the baseline cell. Notably, the 3:1 ratio cell maintained a stable capacity for 100 cycles, unlike the 2:1 ratio cell. However, the cell with the 4:1 ratio showed lower capacity than the 3:1 ratio cell. Figure 5c presents coulombic efficiencies of the cells. The results showed that the TSA-rich cells and baseline cells encountered a distinct decrease at the 37th (1:1), 49th (1:2), and 34th (1:3) cycles, respectively. The observed decreases are indicative of the crossover phenomenon. This is because the coulombic efficiency, defined as the ratio of discharge capacity to charge capacity, is reduced due to the migration of active species across the membrane, which leads to a corresponding reduction in discharge capacity [33]. Conversely, tiron-A-rich cells exhibited stable efficiency retention over 100 cycles. The energy efficiency results, presented in Figure 5d, showed that the TSA-rich cells initially had higher energy efficiency, but this metric decreased with extended cycling. In contrast, the 3:1 and 4:1 ratio cells showed both sustained energy efficiency and stable capacity retention. Consequently, for the tiron-A/TSA RFB system, a higher volumetric ratio of the tiron-A (catholyte) relative to TSA (anolyte) leads to improved cell performance. This is attributed to the higher energy density of the TSA electrolyte compared to the tiron-A electrolyte. Therefore, balancing the electron capacity between anolyte and catholyte Via volumetric control critically influences both capacity and energy efficiency retention. The voltage profiles for the 1st and 100th cycles were recorded to better understand the cell behavior (Figure 5e,f). Firstly, in Figure 5e, the TSA-rich cells did not show a distinct redox reaction plateau between 1.2 V and 1.3 V during the charge phase at the first cycle. This suggests that the tiron-A was fully oxidized before the cell voltage reached 1.2 V. On the other hand, a charging reaction was observed around 1.28 V for the tiron-A-rich cells. Consistent with the capacity results, the tiron-A-rich cells also showed a higher discharge capacity than the TSA-rich cells. By comparing these results with the 1:1 ratio, the electron imbalance between tiron-A and TSA is obvious, and it is effectively alleviated by increasing the volume of tiron-A. Although the 4:1 ratio cell exhibited a high charge capacity, its discharge capacity decreased relative to the 3:1 ratio. This confirms that the cell capacity is optimized at the 3:1 ratio. In the 100th cycle results (Figure 5f), no clear plateau reaction occurred between 1.2 V and 1.3 V for any of the cells during the charging process, and the characteristic voltage curves generally disappeared due to degradation. However, the 3:1 ratio cell uniquely maintained the characteristic shape of its first-cycle voltage profile even after 100 cycles. According to the results, a unique trend was observed that the 4:1 ratio cell initially exhibited higher energy efficiency and retention compared to the 3:1 ratio cell. (Figure 5d) Hence, the further experiment extending the cycling to 50 cycles was conducted to clarify the long-term optimal performance to examine the corresponding Charge–Discharge energies (Figure 6). The results clearly show that the energy retention and energy density of the 3:1 ratio cell were higher than the 4:1 ratio cell over extended operation. This further supports that the 3:1 ratio is the optimal volumetric balance for the tiron-A/TSA system by stabilizing the crucial redox reactions and ensuring superior long-term energy output and efficiency. To investigate the origin of this optimal volumetric balance, Nyquist plots of tiron-A-rich cells were analyzed, and the results are shown in Figure 7.

According to Figure 7a, the ohmic resistance of the cell before Charge–Discharge cycles was lowest at the 3:1 ratio, while it increased again at the 4:1 ratio. The specific, the ohmic resistance (x-intercept) values were 0.127 Ω for the 1:1 ratio, 0.104 Ω for the 2:1 ratio, 0.093 Ω for the 3:1 ratio, and 0.107 Ω for the 4:1 ratio, respectively. Similarly, Figure 7b showed the same trend after cycling, with values of 0.126 Ω for the 1:1 ratio, 0.115 Ω for the 2:1 ratio, 0.103 Ω for the 3:1 ratio, and 0.111 Ω for the 4:1 ratio, respectively. These consistent results indicate that ohmic resistance is a dominant factor in the tiron-A/TSA RFB system. The equivalent circuit parameters derived from Nyquist plots (Figure 7) for the pristine and post-cycled cells are presented in Table 4. Unlike the Nyquist plot for the electrolyte alone, these parameters represent the specific resistance for each cell component, allowing for a clearer distribution among the electrolyte resistance, the charge transfer resistance, the interfacial resistance, and the resistance due to diffusion. The Ls corresponds to inductive resistance, Rs corresponds to solution (electrolyte) resistance and membrane resistance, R1|Q1 corresponds to charge transfer resistance between electrode (carbon felt) and electrolyte, R2|Q2 corresponds to interfacial resistance, R3|Q3 corresponds to additional interfacial and surface resistance, and W is Warburg impedance [34]. Initially, the Warburg impedance, which represents mass transport limitation, is observed to convert to a third semi-circle (R3|Q3) for every post-cycled unit cell. This indicates that the mass transport limitations of the active species are partially alleviated or transformed during the charge–discharge cycles. The total resistance varied significantly among the ratios, with 2:1 exhibiting the lowest pristine resistance. However, the most considerable finding is the notable decrease in total resistance for all cells after 100 cycles, suggesting an activation phenomenon that cycling improves electrode wettability and interfacial contact [35]. The 3:1 cell achieved the lowest total resistance post-cycling, which indicates its superior long-term performance and stability. The detailed analysis of the resistance components reveals that the primary resistance reduction came from the charge transfer resistance (R1), which decreased by 165.11 Ω in the 3:1 ratio. Moreover, the 3:1 ratio showed the lowest formation of additional resistive elements (R3), with the value of only 0.06 Ω. This result confirms the enhanced chemical stability and minimized degradation of the electrode/electrolyte interface at this specific tiron-A:TSA concentration. The stable Rs values across all cycles also confirm the robustness of the electrolyte and membrane components. Therefore, the 3:1 ratio cell showed optimal performance, showing the best combination of stable capacity, high efficiency retention, optimal ohmic resistance, and the lowest total resistance after long-term cycling.

## 4. Conclusions

In this study, we systematically investigated the optimal supporting electrolyte and volumetric ratio for a high-performance tiron-A/TSA RFB system. The CV and EIS results clearly demonstrated that sulfuric acid is the optimal supporting electrolyte. This is based on its ability to produce a higher EMF of 0.860 V and exhibit lower ohmic resistance and superior diffusion coefficients compared to sodium chloride and sodium hydroxide electrolytes. The volumetric ratio of the tiron-A (catholyte) to the TSA (anolyte) significantly influences cell performance. The cells with a higher proportion of tiron-A exhibited excellent performance due to the inherent electron imbalance. The optimal performance was achieved at the 3:1 tiron-A:TSA ratio cell. This cell demonstrated the highest discharge capacity and the most stable performance, and maintained its characteristic voltage profile. The EIS results confirmed that the 3:1 ratio showed the lowest ohmic resistance both initially and after 100 cycles. This confirms that ohmic resistance is the dominant performance factor. In addition, the 3:1 cell showed the most substantial reduction in charge transfer resistance post-cycling, confirming excellent long-term interfacial stability. These findings offer a promising and quantified approach for developing a high-performance tiron-A/TSA RFB system through precise control of the electrolyte environment and volumetric balancing of reactive species.

## Figures and Tables

**Figure 1 materials-18-04614-f001:**
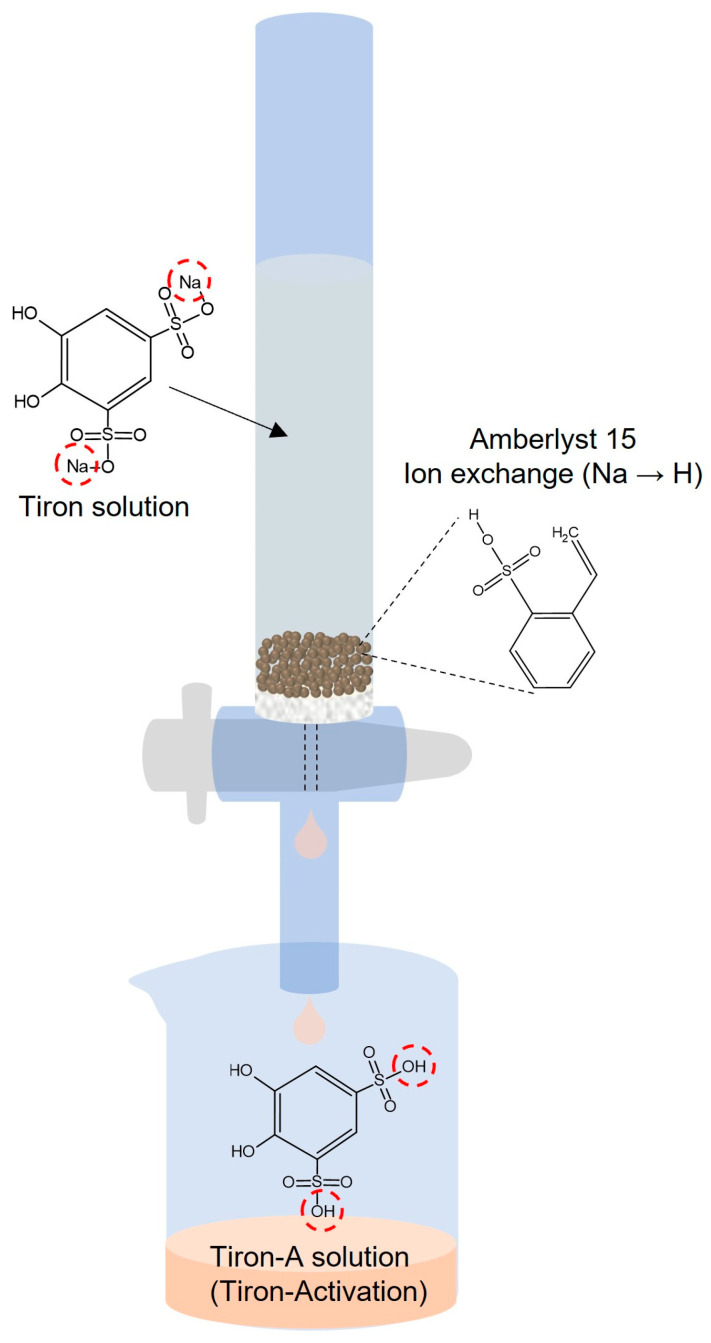
Schematic of the ion exchange mechanism for the tiron-A from the tiron. The Na cation in tiron solution was converted into proton in tiron-A solution by Ambrelyst 15 ion exchange resin.

**Figure 2 materials-18-04614-f002:**
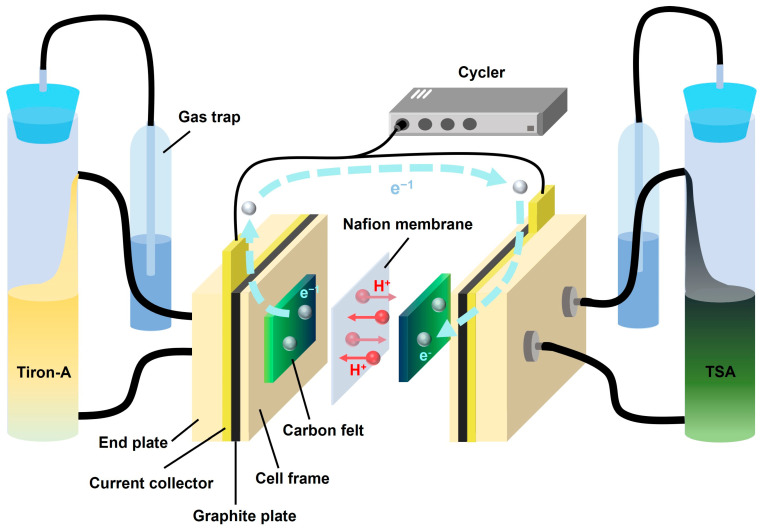
Schematic illustration of the tiron-A/TSA RFB system.

**Figure 3 materials-18-04614-f003:**
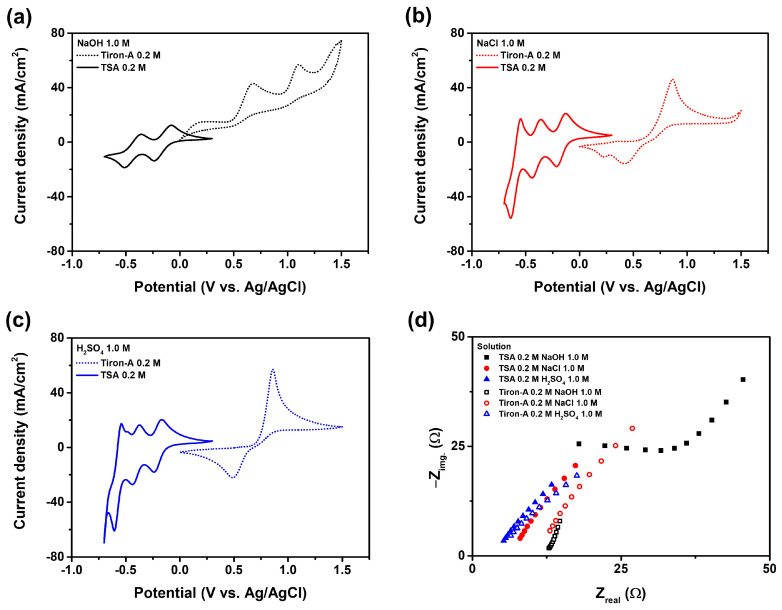
CV and Nyquist plots for tiron-A and TSA in different supporting electrolytes. (**a**) CV for 1.0 M of sodium chloride, (**b**) CV for 1.0 M of sodium hydroxide, (**c**) CV for 1.0 M of sulfuric acid, and (**d**) Nyquist plots for every electrolyte.

**Figure 4 materials-18-04614-f004:**
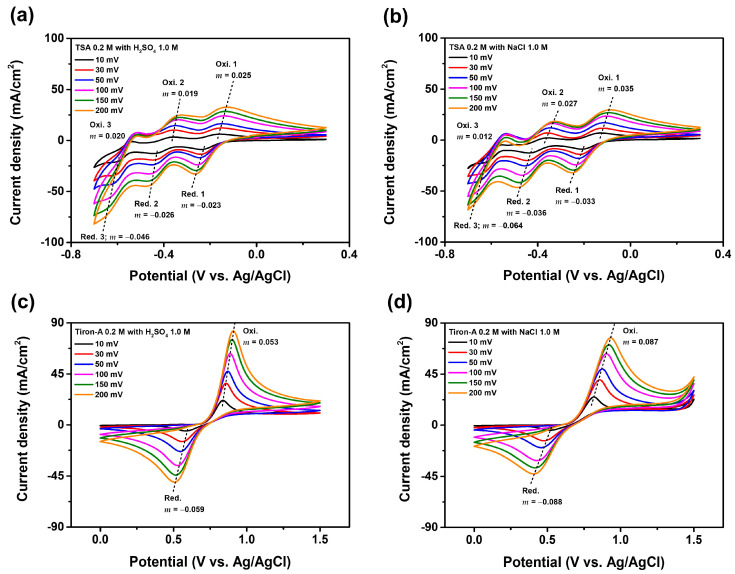
CV results with various scan rates (10 mV, 30 mV, 50 mV, 100 mV, 150 mV, and 200 mV), and the ‘m’ = peak potential vs. log (scan rate) plot slope. (**a**) TSA 0.2 M with H_2_SO_4_ 1.0 M electrolyte, (**b**) TSA 0.2 M with NaCl 1.0 M electrolyte, (**c**) trion-A 0.2 M with H_2_SO_4_ 1.0 M electrolyte, and (**d**) tiron-A 0.2 M with NaCl 1.0 M electrolyte.

**Figure 5 materials-18-04614-f005:**
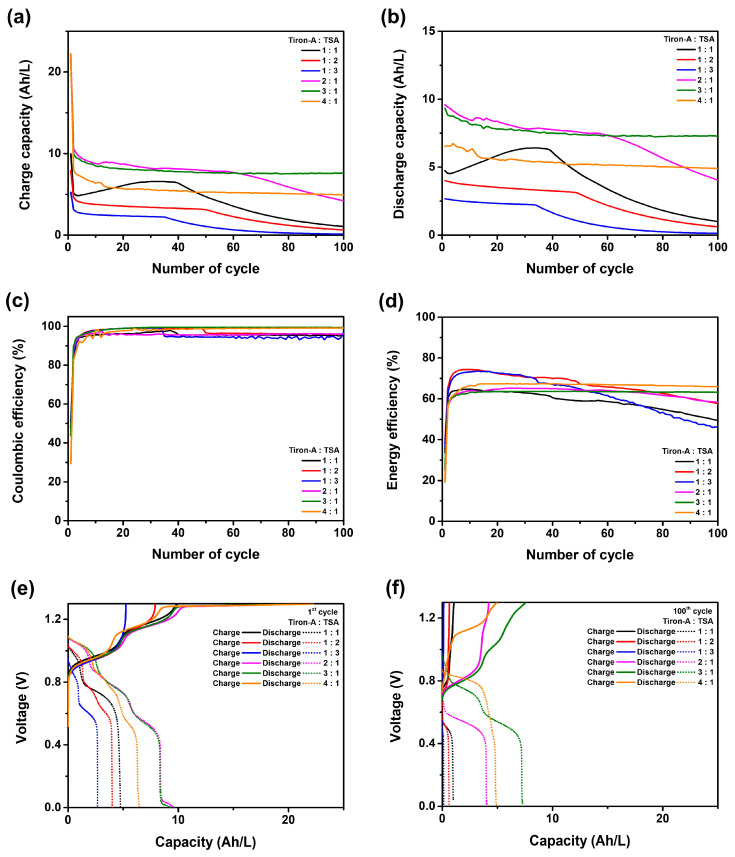
Performance of the unit cells at various volumetric ratios between tiron-A and TSA. (**a**) charge capacities, (**b**) discharge capacities, (**c**) coulombic efficiencies, (**d**) energy efficiencies, (**e**) voltage profiles during the 1st cycle, and (**f**) voltage profiles during the 100th cycle. The pH between TSA and tiron-A is −0.27 and −0.07.

**Figure 6 materials-18-04614-f006:**
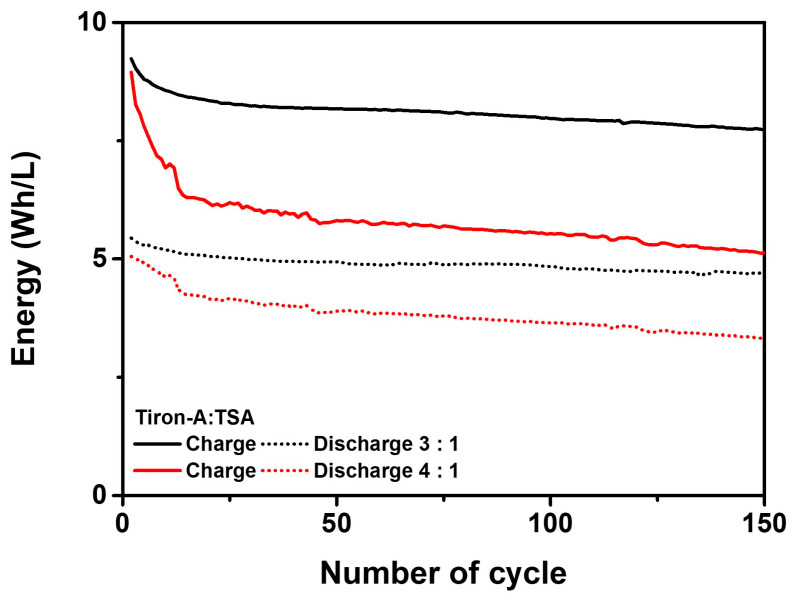
Charge–discharge energy performance for both 3:1 and 4:1 tiron-A:TSA ratio cells.

**Figure 7 materials-18-04614-f007:**
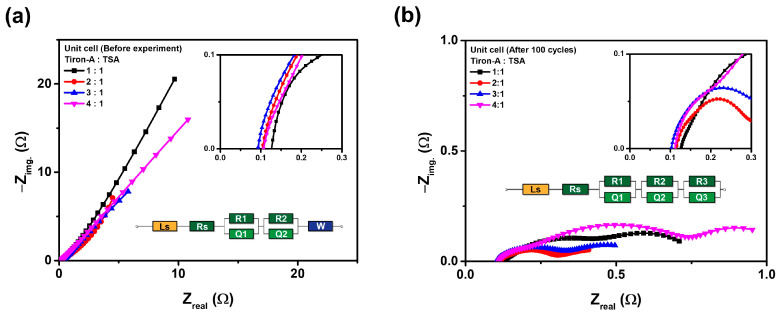
Nyquist plots of unit cells with varying volumetric ratios (1:1, 2:1, 3:1, and 4:1) and their equivalent circuits (**a**) pristine cells, and (**b**) cells after 100 Charge–Discharge cycles.

**Table 1 materials-18-04614-t001:** Respective oxidation and reduction potential values from sodium chloride and sulfuric acid supporting electrolyte.

Reaction	Potential (V vs. RHE)			
	NaCl 1.0 M Electrolyte		H_2_SO_4_ 1.0 M Electrolyte	
	Oxidation	Reduction	Oxidation	Reduction
H(SiW_12_O_40_)^3−^ + e^−^ + H^+^ ↔ H_2_(SiW_12_O_40_)^2−^	0.072	−0.014	0.014	−0.057
H_2_(SiW_12_O_40_)^2−^ + e^−^ + H^+^ ↔ H_3_(SiW_12_O_40_)^−^	−0.162	−0.244	−0.190	−0.255
H_3_(SiW_12_O_40_)^−^ + e^−^ + H^+^ ↔ H_4_(SiW_12_O_40_)	−0.349	−0.442	−0.358	−0.423
(O)_2_^2−^C_6_H_2_(HSO_3_)_2_ + 2e^−^ + 2H^+^ ↔ (OH)_2_C_6_H_2_(HSO_3_)_2_	1.099	0.649	1.055	0.682

**Table 2 materials-18-04614-t002:** The measured pH values with various supporting electrolytes.

Supporting Electrolyte	pH	
	TSA (0.2 M)	Tiron-A (0.2 M)
NaOH 1.0 M	4.63	13.5
NaCl 1.0 M	−0.01	0.58
H_2_SO_4_ 1.0 M	−0.27	−0.07

**Table 3 materials-18-04614-t003:** The respective calculated diffusion coefficients for each peak.

Electrolytes	Diffusion Coefficient (10^−9^ cm^2^/s)					
	Oxi. 1	Oxi. 2	Oxi. 3	Red. 1	Red. 2	Red. 3
TSA 0.2 M with H_2_SO_4_ 1.0 M	2.01	1.25	Non-diffusion	1.73	3.01	8.93
TSA 0.2 M with NaCl 1.0 M	1.55	0.80	Non-diffusion	1.60	3.32	6.52
Tiron-A 0.2 M with H_2_SO_4_ 1.0 M	1.35	-	-	0.74	-	-
Tiron-A 0.2 M with NaCl 1.0 M	0.95	-	-	0.53	-	-

**Table 4 materials-18-04614-t004:** Specific parameters from equivalent circuits for both pristine and post-cycled unit cells.

Parameter	Tiron-A:TSA							
	Pristine Unit Cell (Ω)				Post-Cycled Unit Cell (Ω)			
	1:1	2:1	3:1	4:1	1:1	2:1	3:1	4:1
Rs	0.11	0.08	0.07	0.09	0.11	0.10	0.08	0.10
R1	0.11	58.33	165.33	0.12	0.29	0.14	0.22	0.39
Q1	0.88	2.11	0.82	0.70	2.76	0.76	0.67	0.73
R2	1643.75	0.25	0.36	3553.54	0.16	0.04	0.25	0.28
Q2	0.94	0.68	0.64	0.70	2.19	0.83	3.78	5.27
R3	-	-	-	-	0.28	0.36	0.06	0.11
Q3	-	-	-	-	0.81	5.94	0.87	0.70
W	0.16	0.18	1.02	1.76	-	-	-	-
Total	1645.95	61.63	168.24	3556.91	6.60	8.17	5.93	7.58

## Data Availability

The original contributions presented in this study are included in the article. Further inquiries can be directed to the corresponding author.

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
