# Peer review of "Optimizing Volumetric Ratio and Supporting Electrolyte of Tiron-A/Tungstosilicic Acid Derived Redox Flow Battery"

_materials, 2025, doi:10.3390/ma18194614_

Round 1

Reviewer 1 Report

Comments and Suggestions for Authors

The Authors studied volumetric ratio and supporting Electrolyte of Tiron-A/tungstosilicic Acid Derived Redox Flow Battery.

My comments:

Abstract: Define the acronym. What is TSA?

Abstract: What is Tiron-A? is this a trademark? define its chemical structure.

Line 51: What is the CAS number of tiron-A (2,4,5-dihydroxybenzene-1,3-disulfonate)? 

Line 51: What is the CAS number of tungstosilicic acid (TSA)?

Line 61: give chemical name and cas number of Tiron.

Figure 1: explain to the reader what is Amberlyst 15.

Figure 3: Why 0.2M concentration of Tiron-A and TSA are used?

What is the max solubility limit of Tiron-A?

What is the max solubility limit of TSA?

Table 1: How the Author can be sure the mechanism of chemical reaction is as written in Table 1? Does the Author have supporting data in characterizing the chemical structure (e.g. NMR or other characterization method other than CV)?

Line 153: I believe the order should be Introduction, Experimental, then Results and Discussion, then Conclusion.

Line 170: Add the experimental condition of CV scan rate, and EIS parameters in this paragraph

Line 201: Why Abbreviation section is at the end?

Make the paper easier to read/follow/understand to the readers.

Author Response

1 comment: Define the acronym. What is TSA?
→ The TSA stands for tungstosilicic acid. We added the full name to the manuscript's Abstract section (line 11).

2 comment: What is Tiron-A? is this a trademark? define its chemical structure.
→ The tiron-A is the activated form of tiron. Its correct chemical structure and nomenclature are 4,5-dihydroxybenzene-1,3-disulfonic acid with the formula (OH)2C6H2(HSO3)2. We added this full name to the Abstract (line 10). During the revision process, we identified a naming inconsistency in some parts of the original manuscript (using 2,4,5-dihydroxybenzene-1,3-disulfonic acid). We carefully revised the overall manuscript to ensure the correct nomenclature, 4,5-dihydroxybenzene-1,3-disulfonic acid, used consistently. We confirm that this correction of the chemical name does not alter the study's primary findings, which focus on the electrochemical effects of the supporting electrolyte and the volumetric ratio of tiron-A and TSA.

3 comment: What is the CAS number of tiron-A (2,4,5-dihydroxybenzene-1,3-disulfonate)?
 → We confirm that 2,4,5-dihydroxybenzene-1,3-disulfonate is not the correct structure for tiron-A. The correct structure, 4,5-dihydroxybenzene-1,3-disulfonic acid, is supported by its registered CAS number, 149-46-2 (URL: https://sielc.com/45-dihydroxybenzene-13-disulfonic-acid). Furthermore, we could not find a CAS number for the alternative name (2,4,5-dihydroxybenzene-1,3-disulfonate), supporting our decision to maintain the structure based on 4,5-dihydroxybenzene-1,3-disulfonic acid throughout the revised manuscript.

4 comment: What is the CAS number of tungstosilicic acid (TSA)?
→ The CAS number for tungstosilicic acid is confirmed to be 12027-43-9.

5 comment: give chemical name and cas number of Tiron.
→ The starting material, Tiron, is accurately defined as disodium 4,5-dihydroxybenzene-1,3-disulfonate (CAS number: 149-45-1). We prepared tiron-A from the reagent 4,5-dihydroxy-1,3-benzenedisulfonic acid disodium salt monohydrate (CAS number: 270573-71-2).

6 comment: explain to the reader what is Amberlyst 15.
→We have added more details regarding the function of Amberlyst® 15 in the Introduction section (lines 89 – 91). Amberlyst ®15 is a strong acid cation exchange resin that replaces the sodium (Na+) ions in tiron with hydrogen (H+) ions, converting the salt into the activated acid form (tiron-A). This process is further illustrated in Figure 1.

7 comment: Why 0.2M concentration of Tiron-A and TSA are used?
→ We fixed the concentration for both active species at 0.2 M. This single concentration was chosen to eliminate concentration as a variable and clearly demonstrate the effects of the two primary factors investigated: the supporting electrolyte selection and the volumetric control ratio. The 0.2 M value was confirmed to ensure stable and complete dissolution for both tiron-A and TSA solutions.

8 comment: What is the max solubility limit of Tiron-A?
→ A previous study confirms that the maximum solubility of tiron is 1.5 M when dissolved in 1.0 M sulfuric acid electrolyte. (Reference: https://doi.org/10.1002/er.7948)

9 comment: What is the max solubility limit of TSA?
→ Based on our solubility results (provided in the table below the comment), we determined the maximum solubility of TSA in 1.0 M H2SO4 supporting electrolyte to be 513.30%. This value was calculated from a series of measurements, with the mass of the deposit being 0.4941 g when the fourth solution was removed.

10 comment: How the Author can be sure the mechanism of chemical reaction is as written in Table 1? Does the Author have supporting data in characterizing the chemical structure (e.g. NMR or other characterization method other than CV)?
 → We determined the actual degree of TSA dissociation in the aqueous solution by measuring the pH value. The TSA reagent used has the formula H4(SiW12O40)∙H2O. The measured pH value for the 0.1 M TSA solution was 0.50, which corresponds to a proton (H+) concentration of 0.316 M (calculated by inverting the pH using the formula [H+] = 10-pH). Based on this result, we strongly conclude that the TSA partially dissociates into H(SiW12O40)3- and three protons in the water. These dissociated three protons are consistent with our electrochemical analysis. The one is that the CV initially shows a reduction reaction for TSA. The other is that the Figure 3 displays three distinct and identical reduction peaks under both sodium chloride and sulfuric acid electrolytes. Therefore, our comprehensive analysis supports that the TSA undergoes three typical redox reactions as detailed in Table 1.

11 comment: I believe the order should be Introduction, Experimental, then Results and Discussion, then Conclusion.
 → We corrected the order.

12 comment: Add the experimental condition of CV scan rate, and EIS parameters in this paragraph.
 → We transferred specific experimental details regarding the electrochemical analysis from the figure captions to the Experimental section (line 122 – 123, and 127 – 128).

13 comment: Why Abbreviation section is at the end?
→ We moved the Abbreviation section to be placed after the Abstract section. 

Reviewer 2 Report

Comments and Suggestions for Authors

General comment:

The authors report on a redox flow battery (RFB) using tiron-A (2,4,5-dihydroxybenzene-1,3-disulfonate) as the catholyte and tungstosilicic acid (TSA) as the anolyte. Tiron was transformed into tiron-A through ion-exchange resin treatment to address the imbalance in electrons involved in the redox reactions. The work focuses on the selection of a suitable supporting electrolyte and the optimization of volumetric ratios between anolyte and catholyte. The results demonstrate that sulfuric acid provides superior performance as a supporting electrolyte and that a 3:1 volumetric ratio of tiron-A:TSA yields the best cycling stability and discharge capacity. Overall, this study does not provide novel insights, as the evaluation of supporting electrolytes and volumetric ratios represents standard procedures in RFB research. Nevertheless, the work may still be of interest to researchers seeking data on tiron-A/TSA systems.

The following comments should be addressed before further consideration.

Comment 1. In the Introduction (Line 54), the authors state: “This is because the energy densities of the anolyte (TSA) and catholyte (tiron-A) differ, since the TSA redox reaction involves four electrons while the tiron-A reaction involves two.” Later, in Line 63, they state: “Moreover, the tiron was transformed into tiron-A through ion exchange resin treatment in this study. This treatment adds two more electrons to manage desirable redox reaction. [21]”. This wording is somewhat misleading, as it appears to suggest that tiron-A intrinsically undergoes a 2-electron process, whereas in fact tiron (2e⁻) is converted into tiron-A (4e⁻) via ion-exchange treatment. The manuscript would therefore benefit from clearer and more consistent descriptions of the redox mechanisms of tiron, tiron-A, and TSA, particularly regarding the transition from the 2-electron to the 4-electron process. In addition, labeling the compounds directly in Figure 1 would help readers better understand the mechanism involved during the ion-exchange resin treatment.

Comment 2. The authors identify two important design factors for RFB systems: (i) the choice of supporting electrolyte and (ii) the optimization of volumetric ratios between anolyte and catholyte. It is not entirely clear from the manuscript whether these aspects have already been emphasized as critical parameters in previous RFB studies. A short discussion of earlier work on electrolyte selection and volumetric balancing would strengthen the context and highlight how this study advances beyond existing literature.

Comment 3. In the Introduction (Line 39), the authors state: “Nevertheless, the VRFB system suffered from energy density limitation.” While this is true, one of the major challenges of VRFB technology is also its high cost, particularly due to vanadium and system components. It would strengthen the Introduction if the authors briefly acknowledged cost as a key limitation in addition to energy density.

Comment 4. In the Materials section (Line 159), the authors write: “The resin was first fixed in a burette and then washed once with a 0.5 M sulfuric acid solution. Subsequently, the tiron solution was poured onto the washed resin.” This description relates to experimental procedure rather than materials. Please move it to the Experimental (Methods) section for clarity and consistency.

Comment 5. As mentioned in Comment 4, the description of the N-115 membrane pretreatment (“The N-115 membrane was pretreated to improve its ionic conductivity using the following sequential method: 1) immersion in 5 wt% hydrogen peroxide for 1 h at 80 °C, 2) washing with DI water for 1 h at 80 °C, 3) immersion in 0.5 M sulfuric acid for 1 h at 80 °C, and 4) a final wash with DI water for 1 h at 80 °C”) is experimental in nature. This procedural information should be placed in the Experimental (Methods) section rather than Materials.

Comment 6. The current structure of the manuscript places the Experimental section after the Results and before the Conclusion. This arrangement makes it difficult for readers to follow the flow of the study, since understanding the methods is essential for interpreting the results. For clarity, the Experimental section should appear either immediately after the Introduction or after the conclusions, rather then between discussion and conclusions.

Comment 7. Regarding the Electrochemical Analysis and Battery Performance experiments, it is not clear whether the measurements were conducted under air or an inert atmosphere. Typically, RFB experiments are performed under N₂ to prevent oxygen interference with redox reactions. The authors should clarify the experimental conditions and discuss any potential effects of performing the experiments in air, if applicable.

Comment 8. There appears to be a discrepancy in the figure order: Figure 2 is presented before Figure 3, but in the text, Figure 3 is referenced before Figure 2. The authors should correct the figure numbering or adjust the text to ensure consistency between the figures and their citations.

Comment 9. In Line 89, the authors state: “The Figure 2-4 shows Nyquist plots for each of the electrolytes. For electrolyte Nyquist plots, the ohmic resistance value, which represents the initial resistance caused by the electrolyte, is particularly important. This value commonly corresponds to the x-intercept of the plot. [24]” However, this is incorrect: Figure 2 depicts the schematic illustration of the tiron-A/TSA RFB system, and Figure 4 shows the performance of the unit cells at various volumetric ratios. The authors should correct the figure references to accurately match the content being described.

Comment 10. In the caption of Figure 4 (“Performance of the unit cells at various volumetric ratios between tiron-A and TSA. 1. Discharge capacity performances, 2. Energy efficiency performances, 3. Voltage profiles during the 1st cycle, and 4. Voltage profiles during 100th cycle. Measurement conditions, 0.0 – 1.3 V cut-off voltage with constant current flow of 360 mA”), it is not specified at which pH these measurements were performed. Since different pH where used, the authors should clarify the pH conditions used for the experiments presented in Figure 4.

Comment 11. From Figure 4-2, it is observed that all tests show relatively low energy efficiency, with values below 80%. The authors should comment on the possible reasons for these low efficiencies, such as ohmic losses, crossover, or kinetic limitations, and discuss whether these values are typical for this type of RFB system.

Comment 12. The authors state that they investigated the optimal supporting electrolyte. However, evaluating the supporting electrolyte is a fundamental first step in any redox flow battery study. The authors should clarify that this assessment is a necessary preliminary step rather than a novel investigation, to avoid overstating its significance.

Comment 13. Abbreviations should be defined at their first appearance in the text. Please provide the full forms for all abbreviations used to ensure clarity for the reader. There are several abbreviations in the abstract and in the main article that are not mentioned with the first appearance in the text.

Author Response

1 Comment 1. In the Introduction (Line 54), the authors state: “This is because the energy densities of the anolyte (TSA) and catholyte (tiron-A) differ, since the TSA redox reaction involves four electrons while the tiron-A reaction involves two.” Later, in Line 63, they state: “Moreover, the tiron was transformed into tiron-A through ion exchange resin treatment in this study. This treatment adds two more electrons to manage desirable redox reaction. [21]”. This wording is somewhat misleading, as it appears to suggest that tiron-A intrinsically undergoes a 2-electron process, whereas in fact tiron (2e⁻) is converted into tiron-A (4e⁻) via ion-exchange treatment. The manuscript would therefore benefit from clearer and more consistent descriptions of the redox mechanisms of tiron, tiron-A, and TSA, particularly regarding the transition from the 2-electron to the 4-electron process. In addition, labeling the compounds directly in Figure 1 would help readers better understand the mechanism involved during the ion-exchange resin treatment.
→ We apologize for the ambiguous explanation in the original Introduction. We confirm that tiron-A undergoes a two-electron process, as clearly demonstrated in Table 1 (lines 83-85), not a four-electron process. We have removed the misleading sentence regarding the addition of two electrons ("This treatment adds two more electrons to manage desirable redox reaction. [21]") and have carefully revised the entire manuscript to ensure the correct two-electron mechanism is stated consistently. Furthermore, we have updated Figure 1 and the corresponding text to accurately specify the information regarding the tiron-A preparation process (line 89 – 91).

Comment 2. The authors identify two important design factors for RFB systems: (i) the choice of supporting electrolyte and (ii) the optimization of volumetric ratios between anolyte and catholyte. It is not entirely clear from the manuscript whether these aspects have already been emphasized as critical parameters in previous RFB studies. A short discussion of earlier work on electrolyte selection and volumetric balancing would strengthen the context and highlight how this study advances beyond existing literature.
→ Thank you for the valuable suggestion. We have added a short but focused discussion on previous studies related to supporting electrolyte selection and volumetric balancing in the Introduction section (Lines 70 – 84). This provides necessary context and strengthens our experimental design. 

Comment 3. In the Introduction (Line 39), the authors state: “Nevertheless, the VRFB system suffered from energy density limitation.” While this is true, one of the major challenges of VRFB technology is also its high cost, particularly due to vanadium and system components. It would strengthen the Introduction if the authors briefly acknowledged cost as a key limitation in addition to energy density.
→ We have incorporated additional limitation factors (Lines 54 – 55) for the all-vanadium RFB (VRFB) system into the Introduction, further justifying the investigation into high-energy density alternatives like POMs.

Comment 4. In the Materials section (Line 159), the authors write: “The resin was first fixed in a burette and then washed once with a 0.5 M sulfuric acid solution. Subsequently, the tiron solution was poured onto the washed resin.” This description relates to experimental procedure rather than materials. Please move it to the Experimental (Methods) section for clarity and consistency.
→ To improve clarity and structure, we have created a dedicated "Preparation" section (Lines 106 – 114) following the Materials section. The tiron-A preparation methods have been moved to this new section (Lines 107 – 110) to present the experimental procedures more clearly.

Comment 5. As mentioned in Comment 4, the description of the N-115 membrane pretreatment (“The N-115 membrane was pretreated to improve its ionic conductivity using the following sequential method: 1) immersion in 5 wt% hydrogen peroxide for 1 h at 80 °C, 2) washing with DI water for 1 h at 80 °C, 3) immersion in 0.5 M sulfuric acid for 1 h at 80 °C, and 4) a final wash with DI water for 1 h at 80 °C”) is experimental in nature. This procedural information should be placed in the Experimental (Methods) section rather than Materials.
→ We have also moved the Nafion N-115 pretreatment methods into the newly established Preparation section (Lines 111 – 114).

Comment 6. The current structure of the manuscript places the Experimental section after the Results and before the Conclusion. This arrangement makes it difficult for readers to follow the flow of the study, since understanding the methods is essential for interpreting the results. For clarity, the Experimental section should appear either immediately after the Introduction or after the conclusions, rather then between discussion and conclusions.
→ We corrected the order.

Comment 7. Regarding the Electrochemical Analysis and Battery Performance experiments, it is not clear whether the measurements were conducted under air or an inert atmosphere. Typically, RFB experiments are performed under N₂ to prevent oxygen interference with redox reactions. The authors should clarify the experimental conditions and discuss any potential effects of performing the experiments in air, if applicable.
→ We confirm that all experiments were conducted in a closed system. However, due to equipment limitations, nitrogen gas was not injected to strictly prevent oxygen ingress. We have revised Figure 2 to provide a clearer and more accurate representation of the actual experimental environment and have mentioned the closed-system condition in the Experimental section.

Comment 8. There appears to be a discrepancy in the figure order: Figure 2 is presented before Figure 3, but in the text, Figure 3 is referenced before Figure 2. The authors should correct the figure numbering or adjust the text to ensure consistency between the figures and their citations.
→ We have checked and corrected the correspondence and references to all figures throughout the manuscript.

Comment 9. In Line 89, the authors state: “The Figure 2-4 shows Nyquist plots for each of the electrolytes. For electrolyte Nyquist plots, the ohmic resistance value, which represents the initial resistance caused by the electrolyte, is particularly important. This value commonly corresponds to the x-intercept of the plot. [24]” However, this is incorrect: Figure 2 depicts the schematic illustration of the tiron-A/TSA RFB system, and Figure 4 shows the performance of the unit cells at various volumetric ratios. The authors should correct the figure references to accurately match the content being described.
→ We have checked and corrected the correspondence and references to all figures throughout the manuscript.

Comment 10. In the caption of Figure 4 (“Performance of the unit cells at various volumetric ratios between tiron-A and TSA. 1. Discharge capacity performances, 2. Energy efficiency performances, 3. Voltage profiles during the 1st cycle, and 4. Voltage profiles during 100th cycle. Measurement conditions, 0.0 – 1.3 V cut-off voltage with constant current flow of 360 mA”), it is not specified at which pH these measurements were performed. Since different pH where used, the authors should clarify the pH conditions used for the experiments presented in Figure 4.
→ We have incorporated the pH measurement results (below) as Table 2 in the Results and Discussion section (line 154). We also added a reference to this table in the caption of Figure 4.

Comment 11. From Figure 4-2, it is observed that all tests show relatively low energy efficiency, with values below 80%. The authors should comment on the possible reasons for these low efficiencies, such as ohmic losses, crossover, or kinetic limitations, and discuss whether these values are typical for this type of RFB system.
→ We acknowledge the general challenge that Keggin-type polyoxometalates often exhibit low energy efficiency due to irreversible reactions. However, this study on tungstosilicic acid was necessitated by the significant lack of existing research demonstrating its viability as an active species in RFBs. Furthermore, the significance of our research is that the optimized tiron-A/TSA system demonstrated energy efficiency comparable to that of previous aqueous organic RFBs utilizing tiron-A.

Comment 12. The authors state that they investigated the optimal supporting electrolyte. However, evaluating the supporting electrolyte is a fundamental first step in any redox flow battery study. The authors should clarify that this assessment is a necessary preliminary step rather than a novel investigation, to avoid overstating its significance.
→ We agree that optimizing the supporting electrolyte is a standard and fundamental preliminary step in RFB development. While this optimization factor may not be inherently novel, it was necessitated by the lack of existing studies utilizing tungstosilicic acid as an active species. Therefore, the detailed results presented herein provide a crucial and foundational demonstration of the optimal electrolyte environment for the tiron-A/TSA system, enabling future research to develop highly effective RFB system.

Comment 13. Abbreviations should be defined at their first appearance in the text. Please provide the full forms for all abbreviations used to ensure clarity for the reader. There are several abbreviations in the abstract and in the main article that are not mentioned with the first appearance in the text.
→ We have moved the Abbreviation section to be placed after the Abstract section.

Reviewer 3 Report

Comments and Suggestions for Authors

This manuscript presents a systematic investigation into the optimization of supporting electrolytes and volumetric ratios for a tiron-A/tungstosilicic acid (TSA) redox flow battery (RFB) system. The study is well-structured, with a clear experimental design supported by electrochemical characterization. The findings regarding the superiority of sulfuric acid as the supporting electrolyte and the optimal tiron-A:TSA volumetric ratio of 3:1 are supported by cyclic voltammetry (CV), electrochemical impedance spectroscopy (EIS), and cycling performance tests. The topic is relevant to the field of energy storage, and the results provide practical insights for the development of efficient RFB systems. However, several aspects require clarification and enhancement to improve the rigor and impact of the work. Major revisions are recommended to address the following issues:

  1. The paper clearly demonstrates that the 3:1 volumetric ratio yields the best performance, primarily attributed to lower ohmic resistance, but the underlying mechanism could be further elaborated. For example, a more in-depth discussion is needed on why an excess of tiron-A (3:1 ratio) optimally compensates for the difference in electron transfer numbers (2e⁻ for tiron-A vs. 4e⁻ for TSA) and how this specifically mitigates crossover or other capacity decay phenomena, rather than solely attributing it to ohmic resistance. A brief schematic or model illustrating the charge/mass balance at the optimal ratio would significantly enhance the discussion.
  2. The judgment of redox peak "integrity" and the mechanistic support are insufficient. The paper concludes that NaOH is unsuitable based on CV, while NaCl and H₂SO₄ show clear peaks, and provides a table of redox potentials for TSA (three steps) and tiron-A (one step) vs. Ag/AgCl. To strengthen the argument for "peak integrity," it is recommended to supplement the analysis with a verifiable procedure: include a series of scan rates from 10 to 200 mV s⁻¹ to examine the effects of diffusion control and redox capacitance on peak current. The changes in peak separation and shape with scan rate can also help exclude artifacts from charge transfer limitations. For the structural reversibility of TSA and the electron transfer number of tiron-A, cite representative reviews/original literature and provide a brief energy level or protonation equilibrium analysis for support.
  3. Experimental operations and data processing need further standardization. The potential reference should be unified to RHE to facilitate cross-pH comparisons of redox potentials in NaOH, NaCl, and H₂SO₄ electrolytes. The charge-discharge voltage window should be symmetric and consistent with the CV voltage range to avoid missing redox reactions at high potentials. Additionally, control experiments with gradient current densities should be supplemented. For EIS, it is recommended to expand the test range to 10⁶–10⁻² Hz and perform equivalent circuit fitting to decompose RΩ and R_ct.
  4. The use of a Nafion membrane is mentioned, but its role and potential impact on the system (e.g., vanadium crossover is a known issue in VRFBs) are not discussed. Please briefly comment on whether crossover was observed or considered in this system and how volumetric ratio optimization might interact with membrane properties to influence long-term stability.
  5. The evaluation of cycling stability and conclusions need to be more objective. It is recommended to provide physical images of the experimental setup and compare specific metrics such as energy retention, volumetric energy density, and areal power density. Notably, the energy retention of the 4:1 sample in the figures appears higher than that of the 3:1 sample without signs of decay. Extending the cycle number is necessary to draw more objective conclusions.
  6. The reference list is generally adequate but could benefit from including more recent key publications (2023–2025) on advanced RFB electrolytes and capacity decay mechanisms to demonstrate a comprehensive grasp of the latest liter

Author Response

Comment #1. The paper clearly demonstrates that the 3:1 volumetric ratio yields the best performance, primarily attributed to lower ohmic resistance, but the underlying mechanism could be further elaborated. For example, a more in-depth discussion is needed on why an excess of tiron-A (3:1 ratio) optimally compensates for the difference in electron transfer numbers (2e⁻ for tiron-A vs. 4e⁻ for TSA) and how this specifically mitigates crossover or other capacity decay phenomena, rather than solely attributing it to ohmic resistance. A brief schematic or model illustrating the charge/mass balance at the optimal ratio would significantly enhance the discussion.

→ We have incorporated a more detailed and specific discussion on the electron capacity imbalance between the tiron-A (2-electron) catholyte and the TSA (4-electron) anolyte. This discussion (Lines 224 – 227 and 234 – 236) demonstrated the balance the overall cell capacity and justifies the use of volumetric control as a key optimization strategy.

Comment #2. The judgment of redox peak "integrity" and the mechanistic support are insufficient. The paper concludes that NaOH is unsuitable based on CV, while NaCl and H₂SO₄ show clear peaks, and provides a table of redox potentials for TSA (three steps) and tiron-A (one step) vs. Ag/AgCl. To strengthen the argument for "peak integrity," it is recommended to supplement the analysis with a verifiable procedure: include a series of scan rates from 10 to 200 mV s⁻¹ to examine the effects of diffusion control and redox capacitance on peak current. The changes in peak separation and shape with scan rate can also help exclude artifacts from charge transfer limitations. For the structural reversibility of TSA and the electron transfer number of tiron-A, cite representative reviews/original literature and provide a brief energy level or protonation equilibrium analysis for support.

→ We appreciate the suggestion to deepen the electrochemical analysis. We have added the diffusion coefficient data as Table 3 and included CV results at various scan rates in Figure 4. The supporting electrolyte discussion (Lines 167 – 180) now includes a quantitative analysis of the diffusion coefficients. Furthermore, the electrochemical reversibility of the active species in H2SO4 versus NaCl electrolytes has been analyzed using the slope of the Ep vs. log v plots, with the results discussed in Lines 180 – 186. We have also added relevant references ([24, 25]) to clearly support the reaction mechanisms of TSA and tiron-A.

Comment #3. Experimental operations and data processing need further standardization. The potential reference should be unified to RHE to facilitate cross-pH comparisons of redox potentials in NaOH, NaCl, and H₂SO₄ electrolytes. The charge-discharge voltage window should be symmetric and consistent with the CV voltage range to avoid missing redox reactions at high potentials. Additionally, control experiments with gradient current densities should be supplemented. For EIS, it is recommended to expand the test range to 10⁶–10⁻² Hz and perform equivalent circuit fitting to decompose RΩ and R_ct.

→ We recognize the importance of reporting potentials relative to the Reversible Hydrogen Electrode (RHE) for precision. Although we used an Ag/AgCl reference electrode in our experiments, we have successfully converted all specific potential values to the RHE scale using a validated equation (equation below) from the reference paper (ACS Energy Lett. 2020, 5, 1083-1087). The fixed RHE potential values are presented in Table 1 and are discussed in detail (Lines 151–157). Linear Sweep Voltammetry (LSV) analysis was conducted to precisely define the appropriate potential operating window for the TSA solution, as detailed in the Results section. 

The LSV results indicated that TSA did not exhibit oxidation up to 1.5 V but was vigorously reduced below -1.0 V. Specifically, an extremely unstable reduction reaction occurred around the -1.3 region. Furthermore, a 4th reduction peak was observed at -0.775 V (in 1.0 M H2SO4) and -0.930 V (in 1.0 M NaCl), and 1 5th reduction peak was found at -0.931 V (in 1.0 M H2SO4). Therefore, we further conducted CV with extending potential range (-1.3 V to 1.3 V). The CV result showed that the redox reactions of TSA had rapidly disappeared as the reduction-oxidation cycles progressed. This phenomenon indicates that the TSA material is being destroyed or irreversibly transformed due to excessive energy input outside of its stable operating range. Consequently, the operation potential in this study was controlled at -0.7 V to ensure the reversibility of the TSA active species. 

We appreciate the feedback regarding the EIS analysis. However, we disagree that extending the frequency range to 106 Hz, as the inductive reactance (XL) is already observed at 105 Hz. Further experiments to extend the EIS frequency range to 10-2 Hz are not feasible currently due to the equipment conditions and time limitations.

To separate and quantify the individual resistance components within the cell, the Nyquist plots were fitted using an equivalent circuit model via Z-MAN software (ZIVE Lab). These fitted results are compiled in Table 4 of the revised manuscript (the corresponding equivalent circuits are now presented in Figure 7.). The detail comparison of the equivalent circuit parameters is now used to elaborate the discussion in the manuscript (line 262 – 286).

Comment #4. The use of a Nafion membrane is mentioned, but its role and potential impact on the system (e.g., vanadium crossover is a known issue in VRFBs) are not discussed. Please briefly comment on whether crossover was observed or considered in this system and how volumetric ratio optimization might interact with membrane properties to influence long-term stability.

→ We have added the coulombic efficiency (CE) data to Figure 5-3, as CE is a key metric for demonstrating the crossover phenomenon. The discussion in lines 213–218 interprets the CE results, linking the observed efficiency drops in TSA-rich cells directly to membrane crossover and the resulting capacity loss.

Comment #5. The evaluation of cycling stability and conclusions need to be more objective. It is recommended to provide physical images of the experimental setup and compare specific metrics such as energy retention, volumetric energy density, and areal power density. Notably, the energy retention of the 4:1 sample in the figures appears higher than that of the 3:1 sample without signs of decay. Extending the cycle number is necessary to draw more objective conclusions.

→ The quality of Figure 2 (experimental setup) has been enhanced for clearer visualization. Furthermore, we conducted additional experiments on the 3:1 and 4:1 ratio cells, extending the cycle to 50 cycles to accurately assess long-term energy retention. (Since the response deadline is September 30th, we cannot proceed with further cycles.) The results confirm that the 3:1 ratio cell exhibited significantly higher energy density and superior energy retention compared to the 4:1 cell. This new figure has been added, and the results are discussed in detail (Lines 244 – 250), strongly supporting the optimality of the 3:1 ratio.

Comment #6. The reference list is generally adequate but could benefit from including more recent key publications (2023–2025) on advanced RFB electrolytes and capacity decay mechanisms to demonstrate a comprehensive grasp of the latest liter

→ We have added more references and introduced relevant recent studies in the Introduction section (Lines 70 – 84).

Round 2

Reviewer 1 Report

Comments and Suggestions for Authors

The manuscript quality has been significantly improved. 

Reviewer 2 Report

Comments and Suggestions for Authors

All my previous comments have been adequately adressed, and the authors have significantly improved the manuscript. I recomend the article for publication.

Reviewer 3 Report

Comments and Suggestions for Authors

This revised manuscript shows significant improvement compared with the previous version. The authors have carefully addressed all the concerns raised in the initial review.